# Excreta Disgust and Adaptive Use of Ecological Sanitation By-Products: Perspectives of Rural Farmers in Burera District, Rwanda

**DOI:** 10.3390/ijerph20186743

**Published:** 2023-09-12

**Authors:** Celestin Banamwana, David Musoke, Theoneste Ntakirutimana, Esther Buregyeya, John Ssempebwa, Gakenia Wamuyu-Maina, Nazarius M. Tumwesigye

**Affiliations:** 1Department of Environmental Health, College of Medicine and Health Sciences, University of Rwanda, Kigali P.O. Box 3286, Rwanda; 2Department of Disease Control and Environmental Health, School of Public Health, College of Health Sciences, Makerere University, Kampala P.O. Box 7072, Uganda; 3Department of Community Health and Behavioural Sciences, School of Public Health, College of Health Sciences, Makerere University, Kampala P.O. Box 7072, Uganda; 4Department of Epidemiology and Biostatistics, School of Public Health, College of Health Sciences, Makerere University, Kampala P.O. Box 7072, Uganda

**Keywords:** disgust, by-product, Ecosan, excreta, farmers, sanitation, resource, waste

## Abstract

Ecological sanitation (Ecosan) by-products are inherently limited in their potential use as excreta resources. Disgust behind human excreta and derivatives continues to challenge the further use of Ecosan-by products. Although treated excreta, including Ecosan by-products, have gradually been adopted worldwide, diverse perspectives among users hinder their use in agro-practices. This study explored perceptions of the use of Ecosan-by products as relates to the disgust of human excreta among rural farmers in Burera district, Rwanda. A qualitative study was conducted amongst three farmers’ cooperatives using Ecosan by-products. We conducted six focus group discussions (FDGs) comprising a total of 48 participants taking into account the following three themes: core excreta disgust, perceived waste, and perceived resource. Thematic analysis was conducted with similar perspectives identified and grouped under emerging sub-themes. The perspectives regarding disgust elicitors included stigma, eversion, phobia, taboos, and health risks. Ecosan by-products were largely perceived as useful, with most farmers trusting and willing to touch the by-products. Psychosocial barriers to using the by-products continued to slow down the adoption of Ecosan for agricultural options. There is a need for increased awareness to scale up the use of Ecosan coupled with effective treatment practices for the products so as to reverse the psychological barriers resulting from traditional excreta disgust over Ecosan-products of faeces and urine.

## 1. Introduction

Perceptions of human excreta are surrounded by disgust and discourse in social psychology [1]. Such disgust is mainly defined as excreta-related “emotion” that a human develops to the extent of avoidance of excreta from his environment [2]. Excreta disgust is expressed in different forms, including facial expression, distancing away from the product, or developing psychological factors such as nausea, nuisance, and foul odour [2]. According to the universal disgust domains, human body excretions, particularly urine and faeces, are elicitors and sources of pathogens and therefore perceived as infectious substances [3,4]. The differences in excreta disgust within society can arise along with the magnitude of threats posed by excreta-related “emotion” [5].

The human adaptive behaviour of pathogenic disgust is in the context of prevention and control of diseases and is based on technological and cultural evolution [4]. Only technological complexities in the adoption of Ecosan were explained in a previous study using Roger’s model [6]. The adaptive behaviour of Ecosan by-products can be explained by the theory of adaptive excreta disgust of Rozin–Haidt–McCauley (RHM) [7] (Figure 1). From this theoretical model, two paradigms are presented: One argues that fresh excreta are contaminated and always perceived as “wastes” to throw away. Another is constructed on the notion of excreta paradigmatic shift as a “resource” to use after being treated under Ecosan [8,9]. Such treatment consists of sanitizing both urine and faeces and subsequently applying safe products such as manure [9]. Both paradigmatic views promote excreta disposal options in the social system and hence the adoption of diverse sanitation technologies [10]. It has been noted that the conventional pit toilets built on excreta as “waste” promote excreta disposal effects on environmental sanitation compared to Ecosan, which crystalizes excreta as a “resource” [11]. The idea behind Ecosan was to divert urine from faeces with a minimal smell for quick decomposition and microbial die-off, then reuse the Ecosan by-products in gardens [12].

Ecosan by-products as a “resource” have gained popularity in areas with agriculture predominance. Although excreta derivatives such as urine and faecal products from on-site or off-site treatment facilities have to be used under WHO and FAO standards, their use was also influenced by cultural norms, beliefs about nature, and values attributed to excreta [13]. Studies [9,14,15] on the use of excreta derivatives in agriculture in countries such as China, Vietnam, and India have reported more use of faecal sludge from faecal sludge treatment plants and Ecosan systems. In these countries, a faecophilic culture among farmers was attributed to the excreta and their derivatives regardless of the treatment efficiency [9]. Contrary, studies on Ecosan adoption in Ethiopia [16] and Bangladesh [17] highlighted high-level disgust and aversion behind human excreta to the extent of rejection of Ecosan in some communities. In these areas, religious taboos of touching excreta and derivatives before praying were a great sin among Bangladesh Muslims [18]. Such excreta nuisance among members of the community continues to be a key obstacle to scaling up Ecosan countrywide. Studies have widely investigated the use of human excreta and derivatives in agriculture with concerns of pathogenicity. However, these studies did not deeply investigate peoples’ adaptive behaviours about excreta derivatives, such as Ecosan by-products, through the lens of excreta core disgust in the areas where the technology was implemented.

More than a decade ago, Ecosan was introduced in rural areas of Burera, Rwanda, in response to poor sanitation and infertile soil [19]. The same area is stressed by improper ways of excreta disposal, and hence excreta-borne diseases are prevalent, particularly soil-transmitted helminths and protozoa, with a prevalence of 62.7% in the Burera district [20]. Before the introduction of Ecosan technology, human excreta was perceived as waste, locally known as “amazirantoki”, meaning “touching is forbidden” [21]. It was noted that during the implementation of Ecosan, 20% of household beneficiaries started to use the by-products on their farms through behaviour change training [21]. When combined with Ecosan expansion in the area and local demand for fertilizers, users of excreta derivatives were organized in cooperatives. However, there is hardly any evidence about actual perceptions of the use of Ecosan by-products among farmers concerning local traditional disgust of human excreta. There is a need to understand the perceptions among farmers on the use of Ecosan by-products, taking into account the traditional excreta disgust to support the best interventions and proper practices to scale up use in the community.

## 2. Materials and Methods

### 2.1. Study Design and Setting

This was a qualitative study using grounded theory on excreta disgust from a theoretical model of Rozin–Haidt–McCauley (RHM) [7] (Figure 1) under the three main themes: excreta disgust elicitors, adaptive risky perspectives, and adaptive resource perspectives. The model explains the psychological changes that occur with the emerging technology of Ecosan in terms of traditional excreta disgust to the modern adoptive excreta derivatives, notably Ecosan by-products (Figure 2). In addition, the model informs health risks associated with frequent exposure to faecal pathogens, adaptive safety precautions, and safe excreta treatment options for productive sanitation. The study was conducted in the Burera district in three rural sectors of Rugarama, Gahunga, and Cyanika, which have the highest number of Ecosan installations with agriculture-predominant land use practices. Farmers are organized in cooperatives and rely on diverse local fertilizers, including that from human excreta. This district is known to have lava rock soils formed after a series of volcanic eruption activities in the area. Such magmatic lava formation presents challenges for pit latrine construction since the rocky soil was very hard to dig for excreta disposal. Burera district was therefore selected as the implementation site of Ecosan as an adaptive sanitation facility in the area for more than a decade. 

### 2.2. Study Participants

The study purposively selected three cooperatives, namely “Dufatanye”, which was known to use Ecosan by-products; “Coppecetem” which predominantly used urine; locally named “Ecosan by-product one”; and “Dusukure Phast”, which used different fertilizers including Ecosan by-products. Given the gender-sensitive nature of the topic, two gender-specific FGDs, one for males and another for females, were conducted in each participating cooperative found in Rugarama, Gahunga, and Cyanika Sectors.

Each FGD comprised six to ten participants, recruited with the support of a cooperative representative following the size of the cooperative, experience, shared norms and practice, willingness to participate, and being a member of the cooperative for more than six months. The 48 FGD participants were selected using convenient sampling.

### 2.3. Data Collection Procedures

The FGD guide was designed in English and was translated into Kinyarwanda and then back-translated for accuracy. The FDG participants were recruited according to their willingness to participate. Characteristics considered in FGD participants included a period of membership in the cooperative as well as socio-demographic characteristics such as (gender, age, education, farming experience, farming categories, and religion). Before data collection, three research assistants who were graduates of environmental health were trained for three days by the author about excreta disgust elicitors and adaptive for Ecosan by-products as the main components of the discussion guide. During data collection, FGD participants were first presented with a picture of Ecosan by-products and how such excreta derivatives can be sources of risks for disease and a resource for agricultural production. In each FGD, a face-to-face discussion with participants was conducted by three trained research assistants. The ‘U’ shaped seating option was followed to enable research assistants and FGD participants to face each other. One research assistant moderated the discussion and organized the group, the second one took notes, and the third coded the themes emerging from the participants’ discussion. A discussion on the farmers’ opinions and experiences on excreta elicitors and adaptive use of Ecosan by-products continued until there were no new ideas emerging.

### 2.4. Data Management and Analysis

The FGDs were recorded using a voice recorder, transcribed verbatim into typed scripts, translated into the English version, and organized by a coding system. The reliability of the data was checked by comparing the results of two research team members coding the data independently. The Principal Investigator (PI) checked the validity of the model by sharing it with participants who had knowledge about the study topic. In addition, the transcripts were returned to five participants for comments and to check their accuracy with the original information.

Qualitative data were organised with the analytical support of NVIVO (v.11). The participants’ demographic characteristics (gender, age, education, farming experience, and religion) were analysed descriptively and presented in the form of a table. Drawing on the key themes from the RHM model, the themes of excreta disgust elicitors, adaptive risk perspectives, and adaptive resource perspectives were analysed. All their responses were analysed according to data meaning, context, and research question by thematic content analysis. After multiple readings of the transcripts, the codes were summarised in the created sub-themes according to the similar meaning.

## 3. Results

### 3.1. Socio-Demographic Characteristics of FGD Participants

Table 1 shows that the majority of the 48 FGD participants were female (55.4%), aged above 46 years (43.8%), with informal education (51.8%), and Christians by religion (71.4%). In addition, 43.8% of the FGD participants had a farming experience of more than 10 years producing food largely for household consumption (85.7%).

### 3.2. Excreta Disgust and Adaptive Perspectives

Participants presented different views about the use of excreta and their derivatives. According to the participants’ perspectives on the use of Ecosan by-products in agriculture, three main themes emerged: Excreta disgust elicitors, adaptive risks perspectives, and adaptive resources perspectives (Table 2).

#### 3.2.1. Excreta Disgust Elicitors

All participants perceived fresh faeces as a nuisance, not safe, and doubtful to use them. However, the level of willingness to use the faeces was not the same as depicted in sub-themes: aversion, stigma, phobia, harassment, and taboo. The participants’ opinions on excreta were different, with some farmers expressing fear about fresh excreta. Such faeco-phobia extended the stigma, harassment taboos, and manipulation of Ecosan by-products, as indicated in one quote cited below.

“*I always fear Ecosan by-products as the same as fresh excreta as potential infectious products. People have gradually lost their social and cultural values due to the external agents that brought foreign Ecosan toilets to expose people to their excreta as sources of diarrhoea in our locality*”.FGD 16b

Other participants are willing to use excreta derivatives such as urine and feaces when they are protected. Farmers were frequently exposed to faecal matter at the time of their emptying, transport, and application in the gardens without any means of protection, as emphasised by FGD83a:

“*Some people touch excreta without any fear during faecal vault removal without any protection. For them, the so-called “Ecosan by-product two” is considered as a safe product while the treatment is not efficient in some latrines due to the shortage of ash. Therefore, some products look fresh with a bad smell to the nearby environment, and hence attract many flies…*”

#### 3.2.2. Adaptive Risk Perspectives

In compliance with WHO/FAO guidelines on faecal sludge management, there were tolerable limits of microbes for using faecal matter in crop production. It is against this background that most participants argued about the lack of protective equipment and high rate of spread as presented in three subthemes: pathogens, health risks, and disadvantages of the use of Ecosan-by products. Referring to the pathogens, FGD13b highlighted:

“*There is a serious concern of transmission of faecal parasites in Ecosan by-products two. Although; people use ash for treatment, insufficient use, poor treatment of faeces and short time of storage for decomposition contribute to the high quantity of microbes in our environment*”.

The health risk concern was well articulated by FGD42a, who said that:

“*I do not know if Ecosan by-products are universal products to reuse as fertilizers in our area where people suffer excreta borne diseases such as diarrhoea, Amoebiasis, and Ascaris as we do not have any means of protection during physical contact*”.

Concerning the disadvantages of the use of Ecosan by-products, FGD43a articulated that:

“*There is a lack of big containers that can collect enough “Ecosan by-products one”, the few available are gradually stolen as they are not well fixed*”.

#### 3.2.3. Adaptive Resource Perspectives

Most participants perceived “Ecosan by-products one and two” as valuable matters to reuse. The level of willingness to consume food produced using Ecosan by-products was also indicated. The same perceptions were more expressed by FGD64b:

“*It is crazy that people do not learn quickly from others on the use of Ecosan by-products. I know some neighbors who were against us at the start of using such products, joined later our cooperative after increasing the yield productivity of Irish potatoes more than two times at the same area of farms*”.

The willingness to consume food produced using Ecosan by-products was highlighted by participants in cooperatives known for the use of excreta, including Ecosan by-products. According to FGD11b:

“*Ahhhhh!!!! Such practice of excreta use in the farms has become normal here, if you doubt the use in the garden, please go on the field and harvest some maize, for sure, it feels more delicious than the ones produced under other fertilizers. We have completely shifted to the use of human excreta fertilizer after receiving their benefits on the longevity of soil nutritive and the high crop yield such as Irish potatoes and maize*”.

The participants highlighted the diverse ways of obtaining information and their sources about Ecosan by-products. Most of them highlighted that they only receive information from local leaders and sanitation agents when they participate in national community work. According to the FGD12a:

“*Here, we face the paucity of information regarding Ecosan, we only learn more on the use of Ecosan and its products in our cooperatives and non-facilitation of exchanges of our experiences to the other farmers’ cooperatives in our district*”.

Other participants indicated that the local radio had created a conversation platform on sanitation issues in the district. However, there is no regular and/or specific talk on the proper use of Ecosan, particularly in the agricultural aspect. Participants presented their perspectives under different sub-themes, such as the handling of “Ecosan by-products one and two”. Most of them indicated that touching urine, either treated or not, was not a problem as considered a safe by-product. One of the participants, FGD63a, stated:

“*It is no problem to touch urine and I remember… when I was a child, my mammy used to give some fresh hot urine as medication for some diseases. No one can be afraid or ashamed to touch his or her urine or another person’s urine as we all know that urine is safe and medication*”.

With the shortage of mechanised equipment and personal protective equipment, handling by-products was an optional practice. Other participants viewed “by-product two” as not being safe but can be touched when the farming season starts. They touch the by-products with their hands during garden activities and then wash their hands when returning home before eating. FGD11b stated:

“*We know that the treatment is not a hundred percent efficient……. but we need to apply “Ecosan by-products two” and other related faecal products from diverse sources on the garden during the agricultural season, we manipulated them by our naked hands due to lack of gloves and when were our hands after we come back home before eating*”.

## 4. Discussion

In this study, women constituted the majority of the cooperative farmers and therefore were more exposed to the use of Ecosan by-products than men. While men and women both contribute to farming activities, when it is time for the application of fertilisers, women are more involved. With respect to religion, there were no differences between Christians and Muslims concerning adaptive perceptions of Ecosan by-products, but Muslims reported higher disgust of fresh faeces. There were no categorical differences in age, educational level, farming purposes, and farming experiences regarding farmers’ perspectives of Ecosan by-products. Some farmers had an extreme fear of fresh faeces as the key psychological factor of avoidance of use. According to these farmers, such excreta disgust elicitor is expressed in adaptive waste perspectives, despite being treated under Ecosan. In contrast, other farmers perceived such treated excreta as a resource and were willing to use them as fertilisers under local pet names of Ecosan by-products “one and two”.

The psychological elicitors expanded from Ecosan by-product rejection as a source of contagious avoidance. Such disgust behaviours, to the extent of avoidance of reminders of pathogenic fresh excreta, hinder the use of Ecosan-by products as resources in agriculture. This evidence of disgust was in the form of psychological barriers that hinder farmers from using excreta derivatives in agriculture [22]. This is consistent with studies [22] on psychological mindsets of excreta as triggered by the idea of contagious and farmers’ avoidance behavior of physical contact as well as its smell and sight with Ecosan by-products. Such core adaptive disgust was an outcome of the learning experience of the child from the mother on excreta contagious and ways of avoidance [17]. The same perspectives of Ecosan by-products, some farmers’ mindsets of fear and stigma against by-products remain unchangeable [23].

The adaptive waste perspectives consider excreta and their derivatives as wastes to dispose of far away from the human environment. Such perspectives of excreta being “contagious” are translated into the views of human health risks that can lead to death [4]. Such views on excreta are not limited to biological pathogenicity and are contagious, in some instances, bound to the individual culture [2]. Although there were no systematic studies on excreta disgust in different parts of the world, studies [3,24] show that excreta disgust is an outcome of past learning experiences from mother to child and remains a universal perspective and, hence touching excreta is considered as a sin in Muslim societies [17,25]. However, there were conflicting messages on the use of excreta as resources while emerging Ecosan technology in similar communities. In these areas, study findings on the adoption of Ecosan remain inconsistent [6]. Some studies [18,26] show that people have fully adopted Ecosan and wide use, while other studies [17,20] showed partial Ecosan adoption either as a sanitation option or an agricultural option. These socio-cultures and religious taboos associated with Ecosan were also reported in the study conducted in South Africa [27,28]. Contrary to these study findings, the main constraint of willingness to use Ecosan by-products was health risks associated with the use of “contagious” products. Therefore, the use of safe treatment practices and personal protection during manipulation of Ecosan by-products can improve confidence and trust among farmers in use of Ecosan by-products.

The adaptive resource perspective requires both applied safe practice and behavior change on the part of the user. Evidence of a paradigm shift from excreta “waste” to “resource” [28] was a key perceived usefulness aspect. Such values were discussed in studies conducted in Sweden [29] and Vietnam [30], and both correlate with the study on excreta economic values in South India [1]. Both studies used the same qualitative methods with a focus on excreta users’ perceptions but differed from this study from the targeted population and type of excreta products. A shift from disgust of excreta to the use of derived products through mass awareness was found in Malawi [12]. This study is similar to the findings of a study conducted during the Ecosan installation and campaign by sanitation actors in Rwanda. Even though a series of training occurred during the implementation, the lack of awareness raising through different communication channels has slowed down the adoption of Ecosan by-products. However, trust among people sharing the social norms could improve the wide adaptive use of the by-products for sustainable sanitation solutions [31] but requires improving safe treatment practices and continuous training among food producers and consumers.

### Study Limitations and Research Direction

Some participants, particularly females, felt ashamed about talking in public about the topic concerning their body excretions. The trained research assistants followed an interview guide for logical responses in an empathetic and professional manner. In addition, the discussion was made in separate sex groups, but at a certain level, there were limited responses. While this qualitative study explored the farmers’ viewpoints on the use of Ecosan by-products, their perceptions were analysed and interpreted following the adaptive excreta perspective model of Rozin–Haidt–McCauley (RHM) that gave broader insights into the subject matter. The study was limited to the perspectives of farmers who produce food under the use of Ecosan by-products as fertilisers, and therefore, future research involving consumers’ perspectives of using Ecosan by-products is still needed.

## 5. Conclusions

In this study, we presented an RHM model to explain the farmers’ adaptive perspectives behind the use of excreta derivatives, particularly Ecosan by-products “one and two”, in the areas of implementation of Ecosan technology. According to the model, the first category of farmers extended their traditional excreta disgust elicitors over Ecosan by-products to avoid and hence not willing to reuse. The second category of farmers perceived Ecosan by-products as contagious wastes and risks to health to dispose of. The third category included farmers who perceived Ecosan by-products as resources and were willing to use them. Such evidence of perspectives in the traditional lens of excreta wastes besides Ecosan treatment hinders the local adoption of Ecosan and hence the poor sanitation. Moreover, the poor treatment of excreta through Ecosan exacerbates disgust and phobia among farmers from any cross-contact with Ecosan by-products.

Given the increasing burden of excreta disposal in the area and the local pressure of reuse in agriculture, there is a need for proper sanitation interventions to scale up Ecosan with effective product treatment practices. Regular training on the proper use of Ecosan-products can reverse the psychological barriers eminent from traditional excreta disgust over Ecosan by-products. The findings can be used by sanitation actors to advocate for proper education and the provision of personal protective equipment as part of best practices for using Ecosan by-products. The findings can also help to improve the sanitation policy through the development of user guidelines on Ecosan by-products for sustainable sanitation and agriculture. Further studies using the lens of the social cognitive theories evaluating the social and cultural evolution of the adoption of Ecosan by-products can support understanding social behavior and psychosocial factors within society in the framework of the scale-up Ecosan worldwide.

## Figures and Tables

**Figure 1 ijerph-20-06743-f001:**
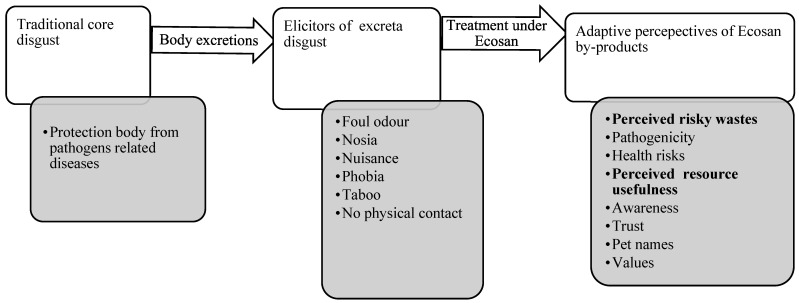
Adaptive excreta disgust model adopted from Rozin–Haidt–McCauley (RHM) framework [7].

**Figure 2 ijerph-20-06743-f002:**
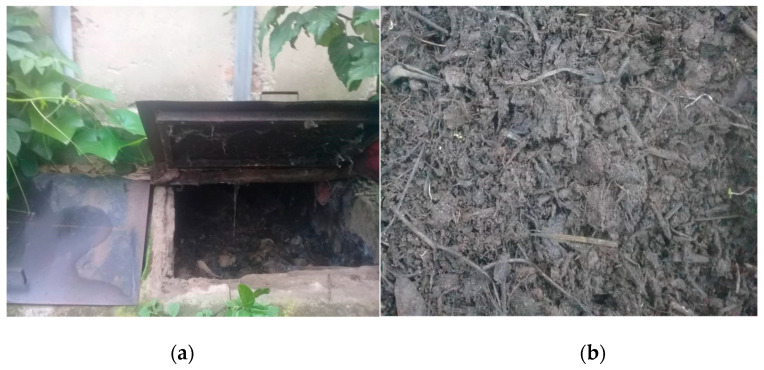
(**a**) Ecosan by-products inside vault. (**b**) A pile of Ecosan by-products.

**Table 1 ijerph-20-06743-t001:** Socio-demographic characteristics of the FGD participants.

Characteristics	Participants (%)
Gender	
Male	21 (44.6)
Female	27 (55.4)
Age(Years)	
18–35	9 (17.8)
36–45	14 (29.4)
Above 46	21 (43.8)
Educational status	
Informal	23 (51.8)
Primary	15 (27.7)
Secondary	10 (20.5)
Religion	
Traditional	12 (25)
Christians	34 (71.4)
Muslims	2 (3.6)
Farming experience( years)	
<5	10 (20.5)
5 to 10	17 (35.7)
>10	21 (43.8)
Farming Purpose	
Subsistence	41 (85.7)
Market	7 (14.3)

**Table 2 ijerph-20-06743-t002:** A comprehensive summary of the resulting sub-theme and codes for each theme.

Themes	Sub-Themes	Codes
Excreta disgust elicitors	Aversion	Bad smell and nuisance, which can attract flies and can transmit fecal pathogens to surrounding environments.
	Stigma	Ashamed to talk about excreta in their local name in public.
	Phobia	Afraid of human feces and do not trust its sanitation.
	Harassments	Transmit fecal pathogens into the environment and local food consumption.
	Taboos	Touching excreta is prohibited.
Adaptive risky perspectives	Pathogenicity	Pathogen content and risks of diseases.
	Health risks	Diarrhoea and amebiasis.
	Disadvantages	Ecosan-products are not sanitary to be expended to all farmers.
Adaptive resource perspectives	Information source	Message on Ecosan during Ecosan installation and sometimes in the national community work.
	Trust and willing	Production of food using Ecosan by-products and the demand is higher.
	Pet names	Urine and faeces are “by-products one and two”, respectively.
	Satisfaction of crop yield	Users are happy with the crop productivity of using both “by-products one and two”.
	Physical contact	“Ecosan by-products two” are dry and sanitary products. There is no fear of manipulating them by hand as other fertilisers.

## Data Availability

All data presented in this study are within the manuscript.

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
