# Peer review of "Excreta Disgust and Adaptive Use of Ecological Sanitation By-Products: Perspectives of Rural Farmers in Burera District, Rwanda"

_ijerph, 2023, doi:10.3390/ijerph20186743_

Round 1

Reviewer 1 Report

In the Data Collection procedures,  row 123, participants were recruited according to their willingness to participate or voluntary participation. you have the wrong word (volunteers).

2.4 Data management, row 143, participants who have knowledge of would be more suited.

3.2.1 191, are you sure you want to use the word afresh? perhaps fresh.

Please check the document and be consistent in the way you refer to the FGD

sometimes it is all one FGD13b  whereas at other times you have a space FGD 42a

On page 7, 244 necked should change to naked

In the Discussion

row 263 humpers should change to hamper or hinder

Row 230

How did you determine the reliability? you stated trust was a reliable factor.

In the conclusion, row 321, fix hamper/ hinder

In rows 327 and 328,  I do not understand your sentence. who is the company? perhaps re-write the sentence and ask for guidance on the English

Check the spacing in lines 83 and 86 when you state the %. space too big

Reviewer 2 Report

Review comments:

Major English proofreading required. Scientific comments are as follows:

1. On what basis the size of the respondents (participants for FGDs) were decided? Is it a fair representation of the entire area ? Include a paragraph in methodology section. Have the author followed the protocol of acquiring the consent of all the adult respondents individually? Why the cooperative representative would give consent for all ? Please justify this.

2. Include a photograph of the eco scan by product & final product for readers.

3. What are the risk assessment methods adopted for safety of people using the treated/untreated excreta by products. Include a paragraph in methodology section.

4. Is there any measures and any hygiene guidelines (from WHO/UN) as health safety and precaution (like sanitation drive/ vaccination) been adopted by authors/researchers to deal with health risks? Include a paragraph in result section.

5. Have the authors considered using statistical tools to plot the results as the current representation of conducting group discussion and reporting gives an impression of a case study and not research. Include some statistical analysis to support the findings.

6. Why everything needs to be man handled? Why mechanised used from treated excreta by-product possible? Include a paragraph in the result section to justify this.

Reviewer 3 Report

Article. Excreta Disgust and Adaptive Use of Ecological Sanitation By- 2 Products: Farmers’ Perspectives in a Rural Area of Burera Dis- 3 trict, Rwanda

Small corrections

Line 83. Please correct a space between the words “62.7%” and “in”.

Line 86. Please correct a space between the words “Ecosan,” and 20%.

Line 117. I believe you need to correct "FGD' to "FGDs".
